# KEYWORD SPOTTER MODEL FOR CROP PEST AND DISEASE MONITORING FROM COMMUNITY RADIO DATA

## ABSTRACT

In societies with well developed internet infrastructure, social media is the leading medium of communication for various social issues especially for breaking news situations. In rural Uganda however, public community radio is still a dominant means for news dissemination. Community radio gives audience to the general public especially to individuals living in rural areas, and thus plays an important role in giving a voice to those living in the broadcast area. It is an avenue for participatory communication and a tool relevant in both economic and social development.This is supported by the rise to ubiquity of mobile phones providing access to phone-in or text-in talk shows. In this paper, we describe an approach to analysing the readily available community radio data with machine learning-based speech keyword spotting techniques. We identify the keywords of interest related to agriculture and build models to automatically identify these keywords from audio streams. Our contribution through these techniques is a cost-efficient and effective way to monitor food security concerns particularly in rural areas. Through keyword spotting and radio talk show analysis, issues such as crop diseases, pests, drought and famine can be captured and fed into an early warning system for stakeholders and policy makers.

## 1 INTRODUCTION

Ensuring a functional and near real-time system of surveillance for crop diseases and pests is of critical importance to sustaining the livelihoods of smallholder farmers in sub-Saharan Africa (Mutembesa et al., 2018). Disease and pest surveillance systems have to be put in place to provide early warning to the farmers and the relevant agricultural research bodies. Usually, when a crop disease or pest is reported in a given area, experts from the respective research institutes take time to reach the reported location to carry out investigations. This usually involves inspection of the crops at specific intervals (of about 10 km) along the more accessible main roads, covering only small proportions of the areas of interest in major districts (Mutembesa et al., 2018). Because the surveillance teams have to work within limited budgets, the surveys and the results from the surveys may be delayed or fewer regions may be sampled in a particular year. As the health experts provide an annual snapshot of the health of crops across the country they are limited in their ability to provide real-time actionable surveillance data. In many cases, the farmers never get to know the disease that has attacked their crops for weeks or even months.

In many areas in Uganda, the vast majority of the affected people will use social media to communicate their concerns in their local communities. This social media is not Facebook or Twitter, its the local community radio stations existing in almost each village in sub-Saharan Africa (Saeb et al., 2017). These rural radio stations give farmers an opportunity to interact with each other and also with the relevant agricultural authorities such as extension workers and agricultural experts. This can usually be through a number of formats like phone-in programs, live talk shows (Nakabugu, 2001). Specifically for some of the radio stations, they have targeted agricultural talk shows that can host an expert from an agricultural research institute who can aim at providing specific information for example: about crop disease and pest management.

Keyword Spotting systems (KWS) is a classification task that aims at detection and retrieving of a series of words from a database of audio streams. The advantage of using a KWS is that unlike full automatic speech recognition systems, they can be developed without sufficient labelled data. This

is common especially in low resourced languages Menon et al. (2019). In this paper, we discuss an implementation of a Keyword Spotting model that we use to mine local community radio content using specific keywords for a low resourced language in Uganda. We evaluate our approach on the Luganda language which is a low-resource language that is currently spoken and used in many of the Agricultural communities in Uganda.

## 2 RELATED WORK

Previous work investigating crop disease surveillance utilizes different approaches. Some approaches have focused on setting up a crop disease surveillance network that relies on the use of mobile phones Mutembesa et al. (2018; 2019) while others use satellite imagery Zhang et al. (2014). The disease detection aspect of the surveillance module uses computer vision and machine learning to detect plant diseases based on leaf imaging Aduwo et al. (2010); Mwebaze & Owomugisha (2016). Leaf-based approaches however rely on the use of imaging device for low-resource approaches utilizing smartphones Quinn et al. (2011); Quinn (2013); Mutembesa et al. (2018; 2019). This may be limited in areas with no or low smartphone adoption.

Keyword spotting for low resource languages has been implemented before Menon et al. (2017); Saeb et al. (2017). The approaches include used include CNNs, siamese CNNs Bromley et al. (1994) and autoencoders. Other models are designed for low computational resources such as Tang & Lin (2018) Coucke et al. (2019) due to the popularity of using keyword spotting to identify commands in smartphones, battery concerns from CPU requirements comes into play. Low resource languages pose a problem as models that consume a lot of data in training fail to converge due to low volume of text corpora and speech recording.

Luganda is an almost zero-resource Bantu language, spoken in the central region of Uganda. Work on Luganda remains small compared to larger languages such as Kiwahili, Zulu and Hausa. Research on Luganda exists in machine translation Nandutu (2016), keyword spotting Menon et al. (2017); Saeb et al. (2017). Prior work on Luganda keyword spotting and radio monitoring of Luganda community radio has been initiated by Menon et al. (2017); Saeb et al. (2017) to generate insights concerning humanitarian aid and development. While radio content is publicly available and accessible with seemingly no data/privacy restrictions, there have been few interventions seeking to mine this data for surveillance purposes particularly for crop pests and diseases.

## 3 METHODOLOGY

### 3.1 BUILDING THE KEYWORD CORPUS

The primary source of keywords in this study are radio recordings captured from radio stations spread across Uganda. We selected 55 radio stations which are commonly listened to in the central region. For each radio station, a google search was done to find out whether it had an online radio station and whether it had its radio schedule available online. From the initial list of radio stations, 19 of them had online streams and of these at least 14 radios broadcast in Luganda. We identified radio schedules for 10 of these radio stations from the radio station websites and also by manually listening in to the stations. The purpose was to identify the time when there are talk shows particularly the agricultural ones. It was observed that for most talk shows topics of discussions are picked depending on the audience demand, the trending topic in the society or country, sponsors/advertisers or specific campaigns though still there are weekly talk shows which are focused on agriculture. The purpose was to identify the time when the talk shows were hired particularly the agricultural ones.

A python script was written to stream the online stations at identified time. These were recorded as 5 minute audio clips which were stored in a shared Dropbox folder. The team also identified 2 radio stations which avail their radio content online for the past 7 days as 1-hour recordings. The websites were scrapped and the audio recordings were sorted depending on whether the 1 hour was a talk show or not. Then the identified 1-hour audio clips that had talk shows were trimmed into 5 minute audio clips and these were added to the shared Dropbox folder.

Table 1: Aggregated Keywords from a sample radio talkshow on the fall army worm pest.

| Sample Keywords | | |
|---|---|---|
| Keyword | Description | Frequency (per recording) |
| Akasanyi | Luganda term meaning *worms* | 18 times |
| Obutunda | Luganda term meaning *passion fruits* | 22 times |
| Kasooli | Luganda word meaning *maize* | 35 times |

The 5-minute audio clips were then played back and carefully listened to by a team of five volunteers with the purpose of identifying and extracting the commonly used agricultural terms that would be fed into the keyword spotting model.

To compliment the keywords captured from radio talkshows, we also scrapped an online local newspaper. For example, we obtained articles from a popular newspaper in Uganda commonly known as Bukedde[1]. One advantage of using online articles as a source of keywords is that there are different ways in which the same crop disease or pest is mentioned and the spelling of such words can be to is captured specifically for Luganda. The keywords were then grouped into crops, diseases, fertilizers, herbicides and general keywords. Translations were also added in 2 languages that is Luganda and English as well as an alternative keyword in form of a stem and this was in case the stem alone was a unique keyword. (give example of keywords with the stemming).

Both the keyword sets from the local radio and online sources were then aggregated into one keyword corpus of 193 keywords. An example of keywords extracted from a radio talkshow on the Fall Army Worm pest affecting maize is shown in Table 1.

## 3.2 SPEECH KEYWORD DATASET

Audio data collection was performed by crowdsourcing speech utterances of the different words from the keyword list. The use of studio captured samples seemed unrealistic, to mimic real-world settings the data was collected in a natural setting with noisy environments, poor quality recording equipment, and people talking in a natural, chatty way. Rather than using high quality microphones, and in a formal setting. This was ensured through the audio data collection tool derived from Warden (2017) where the person speaking out the words can do it using their phone or laptop wherever they are. In this study, we collected data from over 35 users who recorded the keywords in Luganda and English.

An important goal here was to record sufficient data to train the model but low enough to allow for low-resource training. We ensured that we averaged 10 utterances per keyword. Keyword spotting models are much more useful if they are speaker independent, since the process of personalizing a model to an individual requires an intrusive user interface experience. With this in mind, the recording process had to be quick and easy to use, to reduce the number of people who would fail to complete it. The collected keyword audio data was encoded in *ogg vorbis* format.

## 3.3 DATA PREPROCESSING

### 3.3.1 1D CNN

In order to perform speech processing, our first step is to convert the recorded *ogg* keyword files to *wav* files. As ogg is a lossy encoding format, we used ffmpeg ffm to decode the ogg vorbis files into wav audio files. The files were then transformed into 1d vectors using librosa McFee et al. (2019). Then for an audio signal $w_{s_t}$ with a sampling rate $s$ and a length $t$. We use the McFee et al. (2019) resampling feature below to normalize the sampling rate of all the samples to 8kHz as shown in Equation 1.

$$f_{re} : w_{s_t} \rightarrow w_{8kHz_t} \tag{1}$$

---

[1] https://www.bukedde.co.ug

### 3.3.2 SIAMESE CNN

For the Siamese CNN we transform the ogg files into wav using ffmpeg ffm. Then we proceed to generate spectograms and apply mel-compression.

## 4 MODEL ARCHITECTURE AND DESIGN

In this section, we briefly discuss the keyword spotting approaches that we use in this paper.

### 4.1 1D CONVOLUTION MODEL

In this study, We use a 1-dimensional Convolutional Neural Network (CNN) which takes in as input the processed raw audio data. The input to the model is an array representing the audio waveform ($X$). The network is designed to learn the set of parameters ($\theta$) to map the input to a prediction ($T$) according to a hierarchical feature extraction given by equation 3.

$$T = F(X|\Theta) = f_L(...f_2(f_1(X|\theta_1)|\theta_2)|\theta_L) \tag{2}$$

where $L$ is the number of hidden layers in the network.

The final architecture created is a 14-layer deep neural network with five 1D convolutional layers each with intermediate pooling layers. A dropout of 0.5 was also applied after the two successive dense layers. The final layer is a softmax activation function to map to only 10 target keywords selected randomly from the Luganda/English corpus. The model was trained using batch gradient descent with the Adam Optimizer and a learning rate of 0.001.

### 4.2 SIAMESE CNN

The siamese CNN Bromley et al. (1994) takes as an input two mel-spectograms of size 32x100. We generate inputs to the CNN by pairing mel-compressed spectograms of the same word and mel-compressed spectograms of different keywords. The inputs can be 'similar' being pronunciations of the same word or "different" being pronunciations of different words.

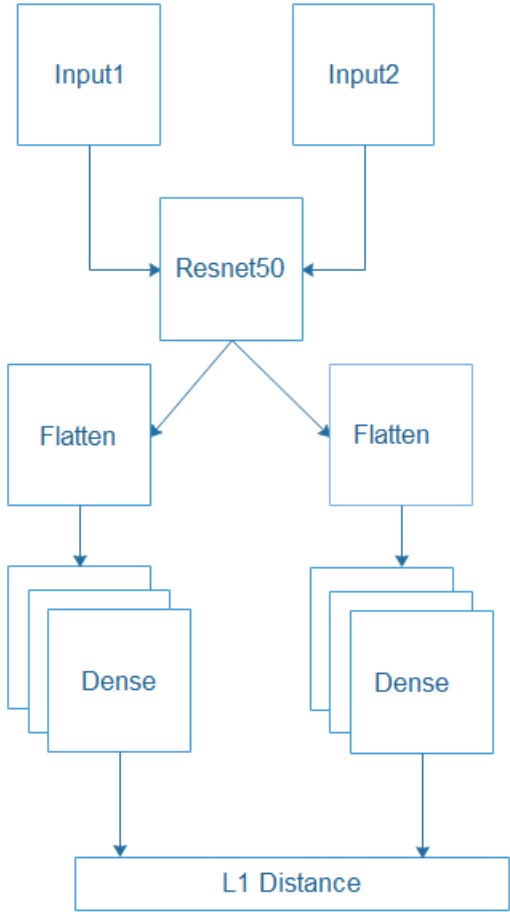

figureSiamese convolutional model

The siamese model performs binary classification at the output, classifying if the inputs are similar or not based on an L1 distance metric. Binary cross-entropy was used as a loss function to achieve this task 3.

$$- \left( y \log(p) + (1 - y) \log(1 - p) \right) \tag{3}$$

## 5   RESULTS AND DISCUSSION

As a baseline experiment we trained a baseline 5-layer densely connected network and our 1d convolutional model. We trained the model with early stopping at no improvement in validation loss for 10 epochs, model stopped at 30 epochs. We trained our model on a total of 18426 samples and validated it on 4607 and tested on 5759 samples. This model was trained on a K80 GPU on Google Colab. The evolution of loss and accuracy are shown in Figure 1. Results are shown in Table 2.

## 6   CONCLUSION AND FUTURE DIRECTION

Using 1D convolutions for low-resource keyword spotting shows promising results. We aim to explore the usability of this work with other Bantu languages, specifically those geographically close to Luganda for example Runyankore, Kinyarwanda and Tooro as well as other linguistically further but geographically close languages such as Luo and Karamojong. We are also interested in the

Table 2: Node Classification Prediction Results

| | Model | Accuracy | Average Precision | Average Recall |
| F-1 | | | | |
| --- | --- | --- | --- | --- |
| Dense Baseline | 0.62 | 0.62 | 0.60 | 0.60 |
| 1d-conv | 0.93 | 0.93 | 0.93 | 0.93 |
| Siamese CNN | 0.85 | 0.92 | 0.75 | 0.83 |

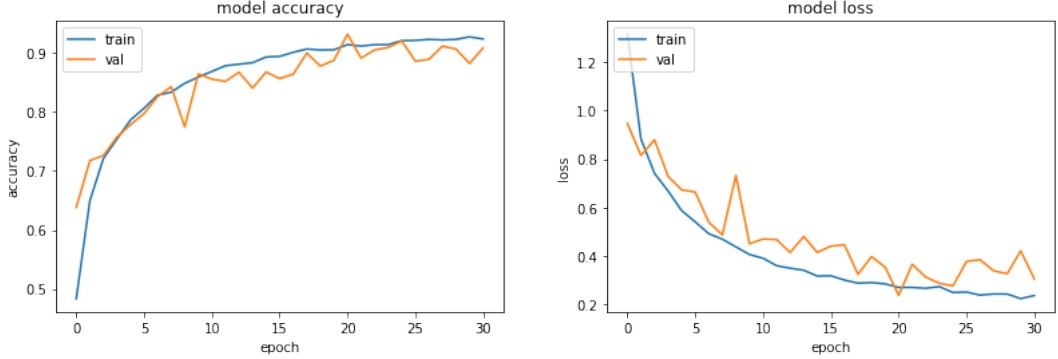

Figure 1: Left: Model training and validation accuracy, Right: training and validation loss.

possibility of using transfer learning in keyword spotting in different dialects of the same language as in Acholi, Lango and Kumam for Luo and different but linguistically related languages as Luganda and Runyankore.

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
