# OpenReview forum: "Keyword Spotter Model for Crop Pest and Disease Monitoring from Community Radio Data"
_ICLR.cc/2020/Conference — Reject_

### Official Review · AnonReviewer3 · 2019-10-15
**Official Blind Review #3**

**Rating:** 1

**Review:**

Overview

This paper presents a very interesting application of speech keyword spotting techniques; the aim is to listen to continuous streams of community radio in Uganda in order to spot keywords of interest related to agriculture to monitor food security concerns in rural areas. The lack of internet infrastructure results in farmers in rural areas using community radio to share concerns related to agriculture. Therefore accurate keyword spotting techniques can potentially help researchers flag up areas of interest. The main engineering challenge that the study deals with is the fact that there isn’t a lot of training data available for languages like Lugandu.

Detailed Comments

 Section 3

1. The word “scrapped” should be replaced with “scraped”.
2. Its not entirely clearly what is meant by “as well as an alternative keyword in form of a stem”. What is meant by “stem” in this context? Maybe a citation or an example would make it more clear.
3. The corpus of keywords supposedly contains 193 distinct keywords however the models in Section 4 are only trained to discriminate between 10 randomly sampled keywords. I don’t understand why this is. Training on the full corpus would allow the network to see more training data and consequently might result in more accurate models.

Section 4

1. The authors refer to Equation 3 in Section 4.1, but I think the reference is to Equation 2.
2. The 1-D CNN has 14 layers but it is not clear to me what the size of the final network is. It would be useful to provide more details of the architecture and a comment about the network size either in terms of number of parameters, number of computations or size of the network weights on disk.
3. The figure for  the Siamese network should be numbered.
4. The authors say that the inputs to the ResNets are of shape 32x100. What do these dimensions refer to? Are their 32 frequency bins? Are their 100 frames as inputs? What are the parameters of the FFT computation? Do 100 frames correspond to a second of audio?
5. How many parameters are their in the Siamese network?
6. Its not clear to me how the Siamese networks are used during inference. The authors say they use the L1 distance to determine if a given test example is similar to a given keyword. How are the examples for the keyword of interest selected? Are all training examples used and the scores average? Are the training examples averaged first to form a centroid vector for  the keyword?

Comments on the methodology

One of the main challenges in this work is the fact that there isn’t a large number of training examples available. However the authors still train relatively large acoustic models with 14 layers for 1-D CNN and a ResNet for the Siamese architecture. There are several studies for keyword spotting on mobile devices that aim to train tiny networks that consume very little power [1] [2]. I think it would be more appropriate to start with architectures similar to these due to the small size of the training dataset. Additionally, it would also be very useful to try and identify languages that are phonetically similar to Lugandu and that have training datasets available for speech recognition. Acoustic models trained on this data can then be adapted to the keyword spotting task using the training set collected for this task. Note that the languages need to have only a small amount of phonetic overlap in order for these acoustic models to be useful starting points for training keyword spotting models.

Comments on the evalation

The keyword spotting models presented here are trained with the intention of applying them to streaming radio data. However, the models are trained and tested on fixed chunks of audio. A big problem with this experimental design is that the models can overfit to confounding audio cues in the training data. For example, if all training data are in the form of 1 second audio chunks and all chunks have at least some silence at the start, then the models learn that a keyword is always preceded by a certain duration of silence. This is not the case when keywords occur in the middle of sentences in streaming audio data. The fact that the evaluation is also performed on individual chunks of audio fails to evaluate how the trained models would behave when presented streaming audio. I am certain that when applied to streaming audio these models would false trigger very often, however this fact isn’t reflected by measuring precision and recall on fixed chunks of test audio.

A more principled evaluation strategy would be to present results in the form of detection-error tradeoff (DET) curves [1], [2]. Here the chunks with examples of the keyword in question can be used to measure the number of false rejects by the system. And long streams of audio data that do not contain the keyword should be used to measure the false alarms. Given that its relatively easy to collect streams of radio data, the false alarms can be measured in terms of the number of false alarms per unit time (minutes, hours or days). This evaluation strategy roughly simulates the streaming conditions in which this model is intended to be deployed. Furthermore, DET curves present the tradeoff between false alarms and false rejects as a function of the operating threshold, which provides much more insight into the performance/accuracy of the trained models.

Summary

I think the area of application of this work is extremely interesting, however the training and evaluation methodologies have to be updated in order to realistically measure the way this system might perform in real-world test conditions.

References

[1] Small-footprint Keyword Spotting Using Deep Neural Networks
[2] Efficient Voice Trigger Detection for Low Resource Hardware


**Experience Assessment:**

I have published in this field for several years.

**Review Assessment: Checking Correctness Of Derivations And Theory:**

I carefully checked the derivations and theory.

**Review Assessment: Checking Correctness Of Experiments:**

I assessed the sensibility of the experiments.

**Review Assessment: Thoroughness In Paper Reading:**

I read the paper thoroughly.

---

### Official Review · AnonReviewer2 · 2019-10-23
**Official Blind Review #2**

**Rating:** 1

**Review:**

This paper tries to design an effective and economical model which spots keywords about pests and disease from community radio data in Luganda and English. The author collected keywords from both radio and online newspapers, set up a Siamese convolutional neural network, which includes input 1 and input 2, Resnet 50, Flatten, five-layer dense and L1 distance, trained the model on a total of 18426 samples and validated on 4607 and tested on 5759 samples via a K80 GPU on Google Colab.

The paper should be rejected because:
(1) The generalizability to different use cases needs to be shown. The approach is too applied.
(2) The fit to ICLR is not perfect. I would expect a stronger focus on representation learning.
(3)	Model
The paper lacks necessary explanation and clarification for the model, which is the core of the paper. For example, the part “Dense” in Siamese convolutional model is not well explained. The number of layers is up to 5, if there is no dropout, the model may have overfitting and gradient vanishing problems.
(4)	Result
As can be seen from both model accuracy and model loss curves in Figure 1, the part of 21st -30th epochs shows an increasing accuracy of training data while that of validation data does not increase anymore and a declining loss of training data while that of validation does not decrease. This is a tendency of overfitting, which means the model should adopt 20 epochs rather than 30 epochs.
(5)	Incompleteness
Compared with the detailed depiction of the first three parts, part 4, part 5 and part 6 are insufficient. It is necessary to complement these three parts with interpretation, explanation and discussion.

Things to improve the paper that did not impact the score:
(1)	Language problems such as in page 3 paragraph 2: “To compliment the keywords captured from radio talkshows, we also scrapped an online local newspaper.”  Here “compliment” should be “complement”, and “scrap” is semantically incorrect.
(2)	Page 4 part 4.1: “…according to a hierarchical feature extraction given by equation 3.” It should be equation 2.
(3)	Format in references: fonts are not uniformed.

**Experience Assessment:**

I have read many papers in this area.

**Review Assessment: Checking Correctness Of Derivations And Theory:**

I did not assess the derivations or theory.

**Review Assessment: Checking Correctness Of Experiments:**

I did not assess the experiments.

**Review Assessment: Thoroughness In Paper Reading:**

I read the paper at least twice and used my best judgement in assessing the paper.

---

### Official Review · AnonReviewer1 · 2019-10-24
**Official Blind Review #1**

**Rating:** 1

**Review:**

The paper describes an approach to analyze radio data with ML-based speech keyword techniques. The authors identify keywords related to agriculture and build a model that can automatically detect these keywords of interest. Their contribution is the proposed model relies on relatively simple neural networks (15-layer 1-D CNNs) to achieve keyword detection of low resourced language (e.g., Luganda). Therefore, they are capable of monitoring food security concerns in rural areas.

This paper should be rejected because (1) no novel algorithm is proposed and only straightforward solution is used (2) the paper never clearly demonstrates the existence of problem they are trying to solve, (3) the experiments are difficult to understand and missing many details, and (4) the paper is incomplete and unpolished.

Main argument
The paper does not do a great job of demonstrating that current model development and drawbacks of dealing with low resourced language (e.g., Luganda). They said they solve the problem by the Keywords Spotting system (KWS), but they did not give a comprehensive review of the current progress of KWS. For example, they should include and compare with the paper, called “Speech recognition and keyword spotting for low-resource languages: Babel Project Research at CUED” by Dr. Mark Gales.

When it comes to methodology, the authors use unnecessary pages to describe how they collect Luganda audios and keyword dataset. It would be appropriate to illustrate their data flow or detection pipeline instead of irrelevant engineering details. In addition, SIAMESE CNN is used to compare performance, but the authors use a relatively large portion of the paper to explain its architecture. Did their method developed beyond SIAMESE CNN? Otherwise, it would be inappropriate to describe it in methodology and model design. For the 1D convolution model, the model is demonstrated in an extremely simple way. They did not even mention their target data (Y), please clarify it in detail.
The experiment did not provide convincing evidence of the utility of the proposed method (1-D convolution). They only demonstrated results in 1 dataset, which did not indicate the effectiveness of the model. There are so many missing details. How many times the experiment they did? How much size is an audio file? Why 1d-conv show the same values of accuracy, precision, and recall? What’s the model drawn in Figure 1? The paper lacks complete discussion on their empirical results. We did not know how to interpret their results.

Considering the above, there are many drawbacks to the paper. The paper seems incomplete. Therefore, we highly recommend rejecting the paper.


**Experience Assessment:**

I do not know much about this area.

**Review Assessment: Checking Correctness Of Derivations And Theory:**

I carefully checked the derivations and theory.

**Review Assessment: Checking Correctness Of Experiments:**

I carefully checked the experiments.

**Review Assessment: Thoroughness In Paper Reading:**

I read the paper thoroughly.

---

### Decision · Program_Chairs · 2019-12-19

**Decision:**

Reject

**Comment:**

Main summary:  Design an effective and economical model which spots keywords about pests and disease from community radio data in Luganda and English.

Discussions:
all reviewers vote on rejecting the paper, due to lack of generalizability, training and evaluation discussion need work
Recommendation: Reject